# Characterization of Serum and Mucosal SARS-CoV-2-Antibodies in HIV-1-Infected Subjects after BNT162b2 mRNA Vaccination or SARS-CoV-2 Infection

**DOI:** 10.3390/v14030651

**Published:** 2022-03-21

**Authors:** Katja G. Schmidt, Ellen G. Harrer, Koray Tascilar, Sabrina Kübel, Boutaina El Kenz, Fabian Hartmann, David Simon, Georg Schett, Krystelle Nganou-Makamdop, Thomas Harrer

**Affiliations:** 1Infectious Diseases and Immunodeficiency Section, Department of Internal Medicine 3, Universitätsklinikum Erlangen, Friedrich-Alexander-Universität Erlangen-Nürnberg, 91054 Erlangen, Germany; katja.g.schmidt@uk-erlangen.de (K.G.S.); ellen.harrer@uk-erlangen.de (E.G.H.); boutaina.elkenz@uk-erlangen.de (B.E.K.); 2Department of Internal Medicine 3, Rheumatology and Immunology, Universitätsklinikum Erlangen, Friedrich-Alexander-Universität Erlangen-Nürnberg, 91054 Erlangen, Germany; koray.tascilar@uk-erlangen.de (K.T.); fabian.hartmann@uk-erlangen.de (F.H.); david.simon@uk-erlangen.de (D.S.); georg.schett@uk-erlangen.de (G.S.); 3Institute of Clinical and Molecular Virology, Universitätsklinikum Erlangen, Friedrich-Alexander-Universität Erlangen-Nürnberg, 91054 Erlangen, Germany; sabrina.kuebel@uk-erlangen.de (S.K.); krystelle.nganou@uk-erlangen.de (K.N.-M.)

**Keywords:** SARS-CoV-2, HIV-1, antibody, humoral immune response, mucosal immunity

## Abstract

Only limited data are available regarding the immunogenicity of the BNT162b2 mRNA vaccine in HIV-1^+^ patients. Therefore, we investigated the humoral immune response after BNT162b2-mRNA vaccination or SARS-CoV-2 infection in HIV-1^+^ patients on antiretroviral therapy compared to HIV-1-uninfected subjects. Serum and saliva samples were analysed by SARS-CoV-2 spike-specific IgG and IgA ELISAs and a surrogate neutralization assay. While all subjects developed anti-spike IgG and IgA and neutralizing antibodies in serum after two doses of BNT162b2 mRNA vaccine, the HIV-1^+^ subjects displayed significantly lower neutralizing capacity and anti-spike IgA in serum compared to HIV-1-uninfected subjects. Serum levels of anti-spike IgG and neutralizing activity were significantly higher in vaccinees compared to SARS-CoV-2 convalescents irrespective of HIV-1 status. Among SARS-CoV-2 convalescents, there was no significant difference in spike-specific antibody response between HIV-1^+^ and uninfected subjects. In saliva, anti-spike IgG and IgA antibodies were detected both in vaccinees and convalescents, albeit at lower frequencies compared to the serum and only rarely with detectable neutralizing activity. In summary, our study demonstrates that the BNT162b2 mRNA vaccine induces SARS-CoV-2-specific antibodies in HIV-1-infected patients on antiretroviral therapy, however, lower vaccine induced neutralization activity indicates a lower functionality of the humoral vaccine response in HIV-1^+^ patients.

## 1. Introduction

Current strategies to control the severe acute respiratory syndrome coronavirus 2 (SARS-CoV-2) pandemic rely on the efficacy of SARS-CoV-2 vaccines. Indeed, in less than one year, several vaccines have been licensed after they had demonstrated high efficacy against development of COVID-19 [1,2,3,4]. The first approved vaccine was BNT162b2 mRNA (Pfizer/BioNTech, COMIRNATY^®^), which is based on a new technology using a modified viral messenger ribonucleic acid (mRNA) encoding the spike (S) glycoprotein of SARS-CoV-2 [1]. Clinical studies in the general population showed a robust generation of SARS-CoV-2-specific antibodies by BNT162b2 mRNA but only limited data are available regarding the humoral immunity in immunocompromised patients [5,6,7]. Over the course of 5 to 10 years, HIV-1 induces a progressive immunodeficiency in most untreated infected patients, which is associated with a poor immune response to vaccination [8]. Modern combination antiretroviral therapy (cART) can suppress HIV-1 replication and improves both CD4^+^ T-cell counts and immunological competence in most compliant patients. Nevertheless, even cART-treated HIV-1-infected patients exhibit generally lower vaccine efficacy as seen for hepatitis B, influenza, and herpes zoster vaccines [9,10,11]. So far, our knowledge regarding SARS-CoV-2-specific immunity in HIV-1-infected subjects on cART remains limited. It has been shown by several groups that neutralizing antibodies play an important role in the control of SARS-CoV-2, both in humans (reviewed in [12,13,14]) and in non-human primate models [15]. Therefore, we investigated the humoral immune response in cART-treated HIV-1-infected subjects and in HIV-1-uninfected subjects after SARS-CoV-2 infection or vaccination with BNT162b2 mRNA. Here, the SARS-CoV-2 spike-specific IgG and IgA antibody levels were measured in serum by ELISA, and the serum neutralizing activity against the S1/RBD domain of the SARS-CoV-2 spike protein was assessed using a surrogate neutralizing antibody assay. As the HIV-1 infection has a negative impact on mucosal immunity [16], we also investigated the SARS-CoV-2 spike-specific IgA and IgG-antibodies and neutralizing activity in saliva. 

## 2. Materials and Methods

### 2.1. Study Subjects

For this study, 82 HIV-1-infected subjects from the Erlangen HIV cohort and 77 HIV-1-uninfected subjects composed of hospital personnel and volunteers from a prospective SARS-CoV-2 seroprevalence study were recruited [17,18]. Inclusion criteria for HIV-1-infected patients were two vaccinations with BNT162b2 mRNA vaccine or past SARS-CoV-2-infection and stable antiretroviral therapy with a viral load of <100 copies/mL. Exclusion criteria were untreated HIV-1-infection, overt clinical disease, a viral load >100 copies/mL and immunosuppressive therapy at the time of sample collection or vaccination/SARS-CoV-2 infection. Inclusion criteria for the controls were two vaccinations with the BNT162b2 mRNA vaccine or past SARS-CoV-2-infection, and absence of HIV-1 infection. Exclusion criteria for healthy controls were overt clinical disease and immunosuppressive therapy at the time of sample collection or vaccination/SARS-CoV-2 infection.

The prime-boost interval between two doses of 30µg of BNT162b2 mRNA vaccine ranged between 20 and 49 days.

The HIV-1-infected subjects were on an effective antiretroviral therapy with a median viral load of <20 copies/mL (IQR: <20 to 20 copies/mL, range <20 to 60 copies/mL). The viral load was suppressed to <20 copies/mL in 51 of 82 subjects. Of the total 159 recruited subjects, 110 subjects were vaccinated with the BNT162b2 mRNA SARS-CoV-2 vaccine, of which 50 were HIV-1-infected and 60 were HIV-1-uninfected.

A total of 43 of the 159 study participants had recovered from asymptomatic to severe COVD-19 infection, of which 26 were HIV-1-infected and 17 were HIV-1-uninfected. Six participants were HIV-1-infected SARS-CoV-2 seronegative subjects that did not receive any SARS-CoV-2 vaccination, did not display any respiratory symptoms at the time of sample collection (Mai 2021) and had no history of COVID-19, thus serving as SARS-CoV-2 non-immune controls. Characteristics of the study participants including gender distributions are summarized in Table 1. Further detailed clinical data for individual subjects are presented in the Appendix A.

The study with analysis of virus specific immune responses was approved by the Ethics Committee of the Medical Faculty (Numbers 250_15B and 157_20 B). Blood and saliva were obtained after informed consent. 

### 2.2. Human Serum and Saliva Sample Collection and Processing

Sera of the 110 BNT162b2 mRNA vaccinated subjects were collected 7–155 days after the second BNT162b2 mRNA vaccination (median of 37 days for HIV-1-infected and 26 days for HIV-1 uninfected). Saliva samples were collected from 34 vaccinated subjects 8–155 days after the second BNT162b2 mRNA vaccination (median of 17 days for HIV-1-infected and 38 days for HIV-1-uninfected). The sera from the 43 SARS-CoV-2 convalescent subjects were collected 7–240 days post PCR diagnosis of COVID-19 disease, as well as saliva samples from 21 convalescents 28–179 days after PCR diagnosis (median of 93 days for HIV-1-infected and 104 days for HIV-1-uninfected). In the four asymptomatic seroconverters with no PCR result (#62, #113, #492 and #586), samples were obtained either 116 and 199 days after the first positive SARS-CoV-2 antibody test or at the first positive SARS-CoV-2 antibody test on day 38 and 264 after the last negative serological test, respectively. For processing of sera, samples underwent a coagulation period of at least 60 min at room temperature prior to centrifugation at 1900× *g* for 10 min at room temperature. All sera samples were stored at −20 °C until further use. The saliva samples were centrifuged at 11,000× *g* for 10 min at room temperature for mucus precipitation and the supernatant was stored at −20 °C until further use.

### 2.3. SARS-CoV-2 IgG-ELISA

The IgG antibodies against the SARS-CoV-2 spike protein were detected using a CE certified commercial enzyme-linked immunosorbent assay (ELISA, Euroimmun, Lübeck, Germany) [19,20,21] according to the manufacturer’s protocol. Briefly, samples were diluted in sample buffer to a final dilution of 1:101 for sera and 1:10 for saliva samples. Per well, 100 µL of calibrator, controls and diluted samples were added in a pre-coated 96 well plate and incubated for 1 h at 37 °C. The wells were washed 3 times with 300 µL washing buffer before addition of 100 µL enzyme conjugate per well. After an incubation period of 30 min at 37 °C, wells were washed 3 times with 300 µL washing buffer. 100 µL substrate was added per well and incubated for 30 min at room temperature. The reaction was stopped with 100 µL stopping solution per well and the absorbance was measured using an ELISA reader (Tecan) at a wavelength of 450 nm with a reference wavelength of 620 nm. The IgG levels were calculated as ratios by dividing the extinction of the sample by that of the calibrator. A ratio of <0.8 was considered negative, a ratio of 0.8–1.1 as borderline and ≥1.1 as positive.

### 2.4. SARS-CoV-2 IgA-ELISA

The IgA antibodies against the SARS-CoV-2 Spike protein were detected using a CE certified commercial ELISA (Euroimmun [21]) according to the manufacturer´s protocol as stated for the IgG ELISA. The IgA levels were calculated as ratios by dividing the extinction of the sample by that of the calibrator. A ratio of <0.8 was considered negative, a ratio of 0.8–1.1 as borderline and ≥1.1 as positive.

### 2.5. SARS-CoV-2 Neutralization Assay

Neutralizing antibodies against the S1/RBD domain of the SARS-CoV-2 spike protein (Wuhan-Hu-1 isolate) were detected using a commercial ELISA (NeutraLISA, Euroimmun) according to the manufacturer’s protocol. Briefly, 20× concentrated ACE2 substrate was diluted to 1× in ACE2 dilution buffer. The sera and saliva samples were then diluted 1:5 in the 1× ACE2 solution. For the 1:101 dilution of sera, the ACE2 substrate was diluted 1:24.75 in ACE2 dilution buffer prior to addition of sera. A total of 100 µL of diluted sample was added per well of a S1/RBD domain-pre-coated 96 well plate and incubated for 1 h at 37 °C. The wells were washed 3 times with 300 µL washing buffer before adding 100 µL enzyme conjugate per well. After an incubation period of 30 min at room temperature, wells were washed 3 times with 300 µL washing buffer. The 100 µL substrates were incubated per well for 15 min at room temperature. The reaction was stopped with 100 µL stopping solution and the absorbance was measured on an ELISA reader (Tecan, Männedorf, Switzerland) at a wavelength of 450 nm with a reference wavelength of 620 nm. The inhibition was calculated as following: $$\%\;\mathrm{IH}=100\%\;-\;\frac{\mathrm{extinction}\;\mathrm{of}\;\mathrm{sample}\;\text{×}100\%}{\mathrm{extinction}\;\mathrm{of}\;\mathrm{blank}\;(\mathrm{mean})}$$

An inhibition of <20% was considered negative, an inhibition between 20–35% as borderline and ≥35% as positive. Negative calculated values (<0%) were considered as 0% inhibition. 

### 2.6. BCA Assay

To determine the protein concentration in saliva, samples were measured using a BCA assay (Pierce™ BCA Protein Assay Kit, Thermo Fisher Scientific, Waltham, MA, USA) according to the manufacturer’s protocol. Briefly, saliva was diluted 1:10 in working reagent. The diluted saliva samples and the BSA standard were incubated in a 96 well microtiter plate at 37 °C for 1 h. After a cooling period to allow the samples to reach room temperature, the absorbance was measured at a wavelength of 560 nm on an ELISA reader. The concentration of the samples was calculated from the determined standard curve. To normalize the IgG and IgA saliva levels, the IgG or IgA ratio was divided by the protein concentration (µg/µL) of the respective sample.

### 2.7. Statistical Analysis

Participant characteristics are presented using appropriate summary statistics. As antibody responses are expected to be affected by age, gender, and time to sample acquisition after exposure to the infectious agent or completion of vaccination, we used linear regression to adjust for these confounders, where age and time variables were centred at the overall mean/median respectively. Model coefficients for the grouping variable, i.e., HIV, control, infected, and vaccinated, indicate the adjusted between-group mean differences and the model intercepts indicate the mean adjusted antibody level in the reference group. In the models for antibody levels between vaccinated groups, a term for median-centred duration between two vaccine doses was also included. All univariate statistical analyses were carried out using GraphPad Prism 9.0.2 (GraphPad Software, San Diego, CA, USA). Multivariate analysis was performed in R v. 4.0.1 software (R Foundation for Statistical Computing, Vienna, Austria). Two-sided *p* values < 0.05 were considered significant without multiplicity adjustment.

## 3. Results

### 3.1. Spike Specific IgG and IgA-Antibodies in Serum after BNT162b2 mRNA Vaccination and COVID-19 in HIV-1-Infected Subjects and HIV-1-Uninfected Controls

For the analysis of the BNT162b2 mRNA induced SARS-CoV-2-specific humoral immune response, we first measured the serum levels of the anti-spike IgG antibodies in 50 vaccinated HIV-1-infected and 60 vaccinated HIV-1-uninfected subjects. After two BNT162b2 mRNA doses, all subjects developed SARS-CoV-2 spike-specific IgG antibodies (*p* < 0.0001). In a univariate analysis, the median spike-specific-IgG antibody level was lower in the HIV-1-infected subjects compared to HIV-1-uninfected subjects (*p* = 0.028, Figure 1a, Appendix A).

Considering possible differences in the kinetics of antibody responses, we found a weak correlation between the serum IgG and IgA levels and time after the second vaccination in both HIV-1 infected and uninfected donors (Appendix A). Therefore, we performed a multivariate linear regression analysis adjusting for age, gender, time after second vaccination and length of the prime-boost interval, which showed no significant difference in the serum spike-specific IgG levels between HIV-1-infected and uninfected persons (*p* = 0.394; Table 2).

Comparison of these vaccine-induced anti-spike IgG levels to those induced after SARS-CoV-2 infection showed significantly higher spike-specific IgG levels in the serum of the BNT162b2 mRNA vaccinees compared to the COVID-19 convalescent subjects, irrespective of HIV status (*p* ≤ 0.0001, Figure 1a, Appendix A), even after adjustment for age, gender, time after second vaccination or time after infection (Table 3). Among COVID-19 convalescent subjects, spike-specific IgG levels were not significantly different between the HIV-1-infected patients and the HIV-1-uninfected subjects (Figure 1a, Appendix A).

In addition to spike-specific IgG antibodies, we also measured spike-specific IgA antibodies in the serum of our study participants. The BNT162b2 mRNA vaccine was able to induce spike-specific IgA antibodies in all vaccinated subjects (*p* < 0.0001, Figure 1b), albeit without significant difference between HIV-1-infected subjects and HIV-1-uninfected controls in a univariate analysis (Figure 1b, Appendix A). Multivariate linear regression analysis adjusting for gender, age, time after second vaccination and prime-boost interval revealed that HIV-1 infection was associated with lower serum IgA-levels (*p* = 0.041, Table 4). 

Serum spike-specific IgA levels did not differ between vaccinees and COVID-19 convalescents. While univariate analysis of the spike-specific IgA antibody levels in serum indicated higher levels in HIV-1-uninfected BNT162b2 mRNA vaccinees compared to HIV-1-uninfected COVID-19 convalescent subjects (Figure 1b, Appendix A), multivariate analysis adjusting for age, gender, time after second vaccination and time after infection, respectively, showed no significant difference between these groups (Table 5). Among COVID-19 convalescent subjects, the spike-specific IgA levels did not differ between the HIV-1-infected subjects and the HIV-1-uninfected controls.

To estimate the relative contribution of the spike-specific IgG and IgA antibodies to SARS-CoV-2 humoral immunity, we next performed paired comparisons of serum spike-specific IgG and IgA levels per participant in our study. Among vaccinated subjects, the spike-specific IgG levels were significantly higher than the spike-specific IgA levels in HIV-1-infected patients and HIV-1-uninfected donors (*p* < 0.0001; Figure 2). Among COVID-19 convalescent subjects, the serum spike-specific IgG and IgA levels were not significantly different in either the HIV-1-infected (*p* = 0.731) or the HIV-1-uninfected (*p* = 0.669) subgroups (Figure 2).

Moreover, the serum anti-spike-IgG antibody levels showed no significant correlation with the serum anti-spike-IgA antibody levels in the vaccinated HIV-1-uninfected subgroups (*r*^2^ = 0.054, *p* = 0.075) and only a weak significant correlation in the HIV-1-infected patients (*r*^2^ = 0.146, *p* = 0.006, Appendix A). Among COVID-19 convalescent subjects, a significant correlation between the serum IgG and serum IgA antibody levels was observed in the HIV-1-infected as well as the HIV-1-uninfected subgroup (HIV: *r*^2^ = 0.496, *p* < 0.0001; HU: *r*^2^ = 0.304, *p* = 0.022, Appendix A).

### 3.2. Spike-Receptor Blocking Antibodies in Serum of BNT162b2 mRNA Vaccinees and Convalescent Subjects

The functional neutralizing activity of vaccine- or COVID-19 induced antibodies was analysed by an ELISA-based surrogate neutralization assay measuring the ability of serum to inhibit the binding of the ACE2 receptor to the SARS-CoV-2 spike receptor binding domain. This assay has been shown to have an equally high accuracy of determining antibody neutralization capacity compared to conventional neutralization assays using SARS-CoV-2 or pseudotyped vesicular stomatitis virus expressing SARS-CoV-2 spike [22,23]. The BNT162b2 mRNA vaccine induced significantly higher neutralizing activity at a 1:5 serum dilution, both in HIV-1-infected and HIV-1-uninfected vaccinees, than the respective COVID-19 convalescent groups (Figure 3a, Appendix A).

Among the BNT162b2 mRNA vaccinated subjects, there was no significant difference in the serum neutralizing activity between HIV-1-infected subjects and HIV-1-uninfected controls at a 1:5 serum dilution, with levels close to 100% suggesting saturated experimental conditions. Further titration of sera from three donors with high or low neutralizing activity revealed that a 1:101 sample dilution might better reveal differences between samples with high neutralizing activity (Figure 3b). At a 1:101 dilution, the serum neutralizing activity of the HIV-1-infected vaccinees was significantly lower compared to the HIV-1-uninfected vaccinated controls (*p* = 0.031, Figure 3c, Appendix A).

We observed a negative correlation between the neutralization capacity at both dilutions and the time after the second BNT162b2 mRNA vaccination in the subgroup of HIV-1-infected (1:5 *r*^2^ = −0.153, *p* = 0.005; 1:101 *r*^2^ = −0.138, *p* = 0.0078; Appendix A) as well as the HIV-1-uninfected subjects (1:5 *r*^2^ = −0.354, *p* < 0.0001; 1:101 *r*^2^ = −0.341, *p* < 0.0001). Furthermore, the serum neutralizing activity at a 1:101 dilution showed a significant but weak negative correlation to the prime-boost interval in the HIV-1-uninfected subjects only (*r*^2^ = −0.201, *p* < 0.0004, Appendix A). Therefore, we performed multivariate analysis adjusting for age, gender, prime-boost interval, and time after second vaccination. This analysis confirmed the significant difference in neutralizing activity between the BNT162b2 mRNA vaccinated and the COVID-19 convalescent subjects (Appendix A). The significantly lower neutralizing activity of the HIV-1-infected patients at the 1:101 dilution compared to the HIV-1-uninfected subjects was also confirmed (*p* = 0.036, Table 6).

Among the BNT162b2 mRNA vaccinated subjects, serum neutralizing activity correlated with spike-specific IgG levels in HIV-1-infected patients both at 1:5 and 1:101 serum dilutions (linear regression analysis: at 1:5: *r*^2^ = 0.603, *p* < 0.0001; at 1:101: *r*^2^ = 0.364, *p* < 0.0001; Appendix A). Among the HIV-1-uninfected subjects, we observed a correlation between neutralization capacity and spike specific IgG levels as well (at 1:5: *r*^2^ = 0.499, *p* < 0.0001; at 1:101: *r*^2^ = 0.238, *p* < 0.0001). A weak correlation between serum neutralizing activity and spike specific IgA levels was observed in the group of the HIV-1-uninfected vaccinees in both dilutions (at 1:5: *r*^2^ = 0.163, *p* = 0.0019, at 1:101: *r*^2^ = 0.188, *p* = 0.0005; Appendix A), whereas in the group of the HIV-1-infected subjects spike specific IgA levels only correlated with the neutralizing capacity at the 1:101 dilution (at 1:5: *r*^2^ = 0.046, *p* = 0.135, at 1:101: *r*^2^ = 0.217, *p* = 0.0007). Among the COVID-19 convalescent subjects, serum neutralizing activity at a 1:5 dilution significantly correlated with spike specific IgG and IgA levels in the HIV-1-infected subjects (HIV IgG: *r*^2^ = 0.758, *p* < 0.0001; HIV IgA: *r*^2^ = 0.431, *p* = 0.0005; Appendix A) and with spike specific IgG levels in the HIV-1-uninfected (HU IgG: *r*^2^ = 0.520, *p* = 0.002; HU IgA: *r*^2^ = 0.067, *p* = 0.332).

Thus, the spike-specific IgA and IgG antibodies were measurable in the BNT162b2 mRNA vaccinated and the COVID-19 convalescent HIV-1-infected patients, but both the serum IgA levels and the neutralizing capacity at a 1:101 dilution were significantly lower compared to the HIV-1-uninfected subjects in a multivariate linear regression analysis.

### 3.3. Spike-Specific Antibodies in Saliva of BNT162b2 mRNA Vaccinees and Convalescent Subjects

Given that the mucous membranes of the nasal and oral cavity are the main entry points of SARS-CoV-2, salivary antibodies could be important contributors to immunity against SARS-CoV-2 infection. We therefore measured spike-specific IgG, spike-specific IgA, and neutralizing antibodies in the saliva of 61 of 159 participants from whom saliva was available. Initial titration experiments with saliva samples of the BNT162b2 mRNA vaccinated, the COVID-19 convalescent and the SARS-CoV-2 non-immune subjects, allowed determination of a 1:10 dilution as the optimal dilution resulting in detectable spike-specific antibody levels with minimal background (Figure 4a). At 1:10 saliva dilution, 5/15 (33%) vaccinated HIV-1-uninfected subjects each had detectable anti-spike IgA and IgG levels in saliva (Table 7). In HIV-1-infected vaccinees, anti-spike IgA was detected in 5/19 (26%) subjects and anti-spike IgG in 2/19 (11%) subjects (Table 7).

Among COVID-19 convalescent subjects, 5/8 (63%) HIV-1-uninfected and 9/13 (69%) HIV-1-infected subjects had detectable anti-spike IgA in saliva, whereas 0/8 (0%) HIV-1-uninfected and 1/13 (8%) HIV-1-infected subjects had detectable anti-spike IgG in saliva. Saliva anti-spike IgA prevalence was higher among COVID-19 convalescent subjects compared to BNT162b2 mRNA vaccinated subjects, but this difference was insignificant (*p* = 0.068). In COVID-19 convalescent subjects, anti-spike IgG prevalence in saliva was lower compared to anti-spike IgA (*p* = 0.031). While these data may reflect differences in mucosal antibody responses between BNT162b2 mRNA vaccinated and COVID-19 convalescent groups, there was no significant difference in the prevalence of anti-spike IgA or IgG antibodies in saliva between the HIV-1-infected and the HIV-1-uninfected subjects. 

Regarding antibody levels, saliva spike-specific IgG levels were significantly higher in the BNT162b2 mRNA vaccinated groups and in the HIV-1-uninfected COVID-19 convalescents compared to the SARS-CoV-2 non-immune control group, but only 8 of 55 vaccinated or convalescent subjects had levels above the positive threshold of 1.1 (Figure 4b, Table 7, Appendix A). Although more subjects had detectable levels of anti-spike IgA in saliva (24 of 55 vaccinated or convalescent subjects), there was only a significant difference in frequencies of IgA in saliva between the HIV-1-infected BNT162b2 mRNA vaccinated and the HIV-1-infected COVID-19 convalescent groups. No significant difference was found between the HIV-1-infected and the HIV-1-uninfected subjects within each group (Figure 4b, Appendix A). To compare saliva antibody levels among persons with detectable levels, data were normalized to saliva total protein content for each subject. Vaccinated HIV-1-uninfected persons showed the highest levels of anti-spike IgG antibodies in saliva (median 1.3) but rare IgG seroprevalence in other groups did not allow statistical comparison of anti-spike IgG levels (Table 8). For IgA, the HIV-1-uninfected persons showed higher median levels within the vaccinated or COVID-19 convalescent subgroups, but these differences did not reach statistical significance (Table 8).

Measurement of neutralizing antibodies by the ELISA-based surrogate neutralization assay required a 1:5 dilution of saliva samples. We were able to detect saliva neutralizing antibodies only in one vaccinated subject, but in none of the COVID-19 convalescent patients. Additionally, 4/21 subjects of the COVID-19 group as well as 5/34 vaccinated donors displayed borderline neutralizing capacities of 20–35% in saliva. However, these levels of neutralizing activity did not significantly differ from those of the SARS-CoV-2 non-immune control group, irrespective of HIV status (Figure 4c).

Overall, the anti-spike IgA and IgG antibodies were distinctly less prevalent in saliva compared to serum of the COVID-19 convalescent and the BNT162b2 mRNA vaccinated groups. Despite detectable saliva anti-spike antibodies among the HIV-1-infected and the HIV-1-uninfected subjects, almost no saliva neutralizing capacity was observed in our cohort.

## 4. Discussion

In this study, we investigated the humoral SARS-CoV-2-specific immune response in HIV-1-infected subjects on antiretroviral therapy with a controlled HIV-1 viremia of ≤60 copies/mL. All HIV-1-infected subjects in this study were able to mount SARS-CoV-2-specific IgG antibody responses after a two-dose BNT162b2 mRNA vaccination. Notably, we observed lower serum anti-spike IgA and neutralizing antibody levels in HIV-1-infected subjects after adjustment for age, gender, prime-boost interval, and time after second vaccine dose, denoting a negative effect of the HIV infection on BNT162b2 mRNA-induced humoral responses.

Our study is in line with several recent reports on HIV-1-infected individuals mounting robust spike-specific IgG antibody responses to vaccination with BNT162b2 mRNA, Moderna mRNA-1273 vaccine and ChAdOx1 nCoV-19 vaccines [5,24,25,26,27,28,29,30] comparable to healthy control groups in the majority of studies. Apart from spike-specific IgG, we were also able to detect spike-specific IgA in the serum of all vaccinated study subjects, demonstrating that the intramuscular application of the BNT162b2 mRNA vaccine can induce both IgA and IgG antibodies in HIV-1-infected subjects and in HIV-1-uninfected controls although at lower levels than IgG. This is in line with reports of other groups describing the induction of spike-specific IgG and IgA antibodies by mRNA vaccines in HIV-1-uninfected subjects, with anti-spike antibody levels being consistently higher in vaccinated persons compared to COVID-19 convalescent and serum IgG levels being higher than IgA levels [31,32,33,34,35]. Here, the neutralizing activity correlated significantly with the spike-specific IgG and IgA titers in the serum of HIV-1-infected and -uninfected donors. Studies from other groups showed that both spike-specific IgG and IgA antibodies exhibit SARS-CoV-2 neutralizing activity [36] but the relative importance of neutralizing IgA antibodies compared to the IgG antibodies for serum neutralization and outcome of SARS-CoV-2 infection is still unclear. Thus, the relative contribution of serum IgA to SARS-CoV-2 immunity remains to be clarified in future studies and may be influenced by its relatively lower half-life compared to IgG [32].

In our study, the assessment of the functional activity of the spike-specific antibodies by a surrogate neutralization assay showed that the vaccine was able to generate potent neutralizing activity in serum of both HIV-1-uninfected and HIV-1-infected subjects. However, the significantly lower neutralizing activity at a 20-fold higher serum dilution in the HIV-1-infected subjects than in the HIV-1-uninfected controls indicates a moderate impairment of the humoral immune response in cART-treated HIV-1 infection as seen with other vaccines [8,9,10,11]. This is unlikely to be explained by depletion of CD4^+^ T-cells in the blood, as all patients in our study showed normal or only moderately decreased CD4^+^ T-cells with a median CD4^+^ T-cells count of 634/µL (range 234–1548) and the CD4^+^ T-cells counts did not correlate to the spike-specific IgG-levels and neutralizing activity in the HIV-1-infected vaccinees. A lack of correlation of spike-antibody titers to CD4 counts in cART treated HIV-1-infected patients was also reported in the study of Frater et al. [29] who investigated the spike-specific antibody titers after vaccination with ChAdOx1 in a similar patient group. Nonetheless, incomplete immune reconstitution is commonly reported in cART-treated HIV-1-infection. Incomplete recovery of the repertoire of CD4^+^ T-cells despite normalization of numbers in the peripheral blood, alterations in the differentiation of CD4^+^ T-cell subsets [37] and ongoing immune activation interfering with an efficient immune reconstitution [38] could all play a role in reduced vaccine efficacy. The duration of cART did not correlate to spike-specific IgG, spike-specific IgA and neutralizing antibody titers in a multivariate analysis including the years on antiretroviral therapy.

Moreover, the prevalence of co-morbidities commonly seen in HIV-1 infection such as diabetes, hepatitis B infection could also influence vaccine efficacy. In our study, a limited number of participants (30 out of 82 HIV-1-infected) had any co-morbidities, and our analysis did not show association with antibody responses. Longitudinal studies are needed to clarify whether lower titers of neutralizing spike-specific antibodies observed in our study will be associated with a more rapid decline of humoral immunity and an increased susceptibility to SARS-CoV-2 infection.

A confounding factor in our study was age, as the spike-specific IgG levels showed a weak negative correlation to age and the HIV-1-infected patients were older than the control subjects. It is known that immune responses generally decrease with older age, and this could lead to lower spike-specific IgG titers. In contrast to spike-specific IgG, we did not observe a correlation of spike-specific IgA and neutralizing antibodies to age. A recent study [39] reported strong negative correlation between age and both SARS-CoV-2-specific IgG and neutralizing antibody levels after mRNA vaccination, but the median age in that study was much higher in the older adult group (79 years) compared to the group of HIV-1-infected subjects in our study (55 years). Furthermore, our multivariate linear regression analysis adjusting for age as well as gender, prime-boost interval, and time after second vaccine dose reveals an effect of HIV-1 infection on serum anti-spike IgA and neutralizing antibody levels but not on anti-spike IgG levels. Our findings are different from other studies reporting similar BNT162b2 mRNA-induced serum anti-spike neutralizing antibody levels between HIV-1-infected persons and healthy controls [5,7,40,41]. Differences in sampling time could account for divergence in findings, with our sampling time up to 7.5 weeks in HIV-1-infected and 9.5 weeks in HIV-1-uninfected as opposed to 1–4 weeks. Of note, recent studies on inactivated SARS-CoV-2 vaccines reported similar SARS-CoV-2-specific neutralizing antibody responses between HIV-infected persons and healthy controls up to 4 weeks after vaccination [42] contrasting with lower SARS-CoV-2-specific neutralizing antibody responses in HIV-infected persons 40 days [43] or up 84 days [44] after vaccination. Thus, a faster decay of antibody responses may account for our observation of lower antibody responses after BNT162b2 mRNA vaccination in HIV-1 infection. Differences in sampling time between groups in our study warrant caution with respect to estimating the precise impact of HIV-1 infection on antibody levels after BNT162b2 mRNA vaccination.

To assess the potency of the BNT162b2 mRNA vaccine we compared the humoral immune response of the vaccinees with that of subjects who had recovered from SARS-CoV-2 infection. Indeed, the BNT162b2 mRNA vaccine induced significantly higher spike specific IgG levels compared to SARS-CoV-2 infection. Importantly, vaccination with the BNT162b2 mRNA also induced significantly more spike-specific neutralizing antibodies in the serum compared to the COVID-19 convalescent individuals. The strong differences in the spike-specific IgG levels and the neutralization activity between the vaccinated subjects and the convalescent subjects was influenced by a shorter observation time after the second BNT162b2 mRNA vaccination compared to the observation time after the recovery from active SARS-CoV-2 infection as SARS-CoV-2 antibodies display a two-phase decay after infection [14,23]. However, even after adjustment for time after the second vaccination or infection, the spike-specific IgG levels were significantly higher in vaccinated subjects than the COVID-19 convalescent subjects. This is also in accordance with other studies reporting higher antibody titers after vaccination than after natural infection in non-hospitalized COVID-19 patients [6,14,34,45,46,47,48].

In the convalescent subjects, no difference was found between the HIV-1-infected and the uninfected individuals regarding spike-specific IgG and IgA levels and with the neutralizing serum activity at a 1:5 dilution. This finding is in accordance with the report by Alrubayyi et al. [49] who described similar spike-specific IgG levels and neutralizing antibody titers in the HIV-1-infected patients on ART in comparison to the uninfected controls. In contrast, lower spike-specific IgG titers and pseudovirus neutralizing antibody titers were reported on a cohort of HIV-1-infected subjects from San Francisco [50]. This difference could be explained by lower CD4 counts in that cohort (median CD4 452/µL versus CD4 of 551/µL in this study and CD4 of 571/µL in Alrubayyi et al. [49]) as the authors described significantly lower spike antibody titers in patients with CD4 < 200/µL versus CD4 > 200/µL [50].

Apart from serum, we could also detect spike-specific antibodies in saliva, albeit at lower frequencies and titers than in the serum. Akin to the study of Becker et al. [31], vaccinated subjects displayed spike-specific IgG-responses similar to IgA-responses in saliva, whereas in saliva of convalescent subjects, spike-specific IgA antibodies were more frequently observed than spike-specific IgG antibodies. This may suggest that the location of the priming of the immune response influences the differentiation of B-cells with induction of a stronger spike-specific IgA response by infection via the mucosal route in contrast to the intramuscular application of the vaccine.

Other groups recently reported mucosal antibodies both after SARS-CoV-2 infection and vaccination. In most of those studies, spike-specific saliva antibodies could be detected at higher frequencies than in our study [31,35,40,51,52,53]. This difference could be influenced by the method of saliva sample collection. In our study, saliva was sampled by spitting into a collection tube, while other studies utilized oral swabs or mouthwash samples [52]. In some reports, the saliva samples were concentrated after a centrifugation step [52,53], treated with additives [35] or heat-inactivated [40]. Furthermore, different serological assays were used for the measurement of antibodies. These differences most likely lead to different sensitivities regarding the detection of saliva antibodies. In line with our study, Azzi et al. showed that the saliva antibody levels are substantially lower than the serum IgG and IgA levels [54]. In this study, neutralizing activity in saliva was only detected at low frequencies.

In summary, our study clearly demonstrates that the BNT162b2 mRNA vaccine induces a robust SARS-CoV2 spike specific humoral immune response both in HIV-1-uninfected subjects and in HIV-1-infected patients on antiretroviral therapy, exceeding the antibody response observed in SARS-CoV-2 infected subjects. Nevertheless, vaccine induced serum neutralization activity and SARS-CoV-2-specific IgA levels were significantly lower in HIV-1-infected patients than in HIV-1-uninfected subjects, indicating a moderately lower functionality of the humoral vaccine response. Longitudinal studies are necessary to investigate whether this moderately lower antibody response in HIV-1-infected subjects will lead to a lower vaccine efficacy over time. Low titers of spike antibodies and low neutralizing activity in saliva in comparison to strong antibody titers and neutralization activity serum could explain the risk of infection of both vaccinated and convalescent subjects despite good protection from symptomatic disease.

## Figures and Tables

**Figure 1 viruses-14-00651-f001:**
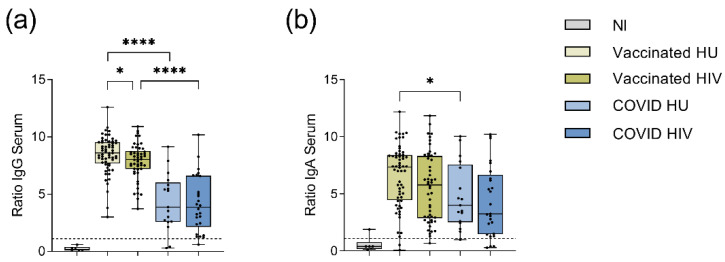
Spike-specific IgG and IgA antibodies in the serum of vaccinated and COVID-19 convalescent subjects as measured by ELISA. The IgG and IgA levels are presented as ratios obtained by dividing the extinction of the sample by that of the assay calibrator. (**a**) Spike-specific IgG and (**b**) Spike-specific IgA in the serum of BNT162b2 mRNA vaccinated (*n* = 110) and COVID-19 convalescent (*n* = 43) subjects. Medians and interquartile ranges (IQRs) are presented for SARS-CoV-2 non-immune (NI, *n* = 6), vaccinated HIV-1-uninfected (HU, *n* = 60), vaccinated HIV-1-infected (HIV, *n* = 50), COVID-19 convalescent HIV-1-uninfected (COVID HU, *n* = 17) and COVID-19 convalescent HIV-1-infected (COVID HIV, *n* = 26). All BNT162b2 mRNA vaccinated and COVID-19 convalescent subjects had significantly higher IgG- (*p* ≤ 0.0001) and IgA-ratios (*p* ≤ 0.001) compared to the NI group (*p* not indicated in the graph). Significance was tested by Mann–Whitney-U test. * = *p* ≤ 0.05, **** = *p* ≤ 0.0001. Exact *p* values are shown in Appendix A. The cut-off ratio for a positive result at ≥1.1 is indicated by dashed lines.

**Figure 2 viruses-14-00651-f002:**
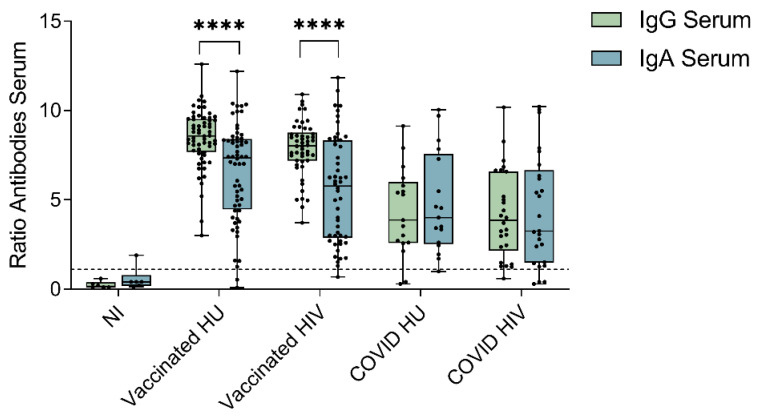
Comparison between the spike-specific IgG- and IgA-levels in serum within BNT162b2 mRNA vaccinated subjects (*n* = 110) and COVID-19 convalescent subjects (*n* = 43). Medians and interquartile ranges (IQRs) are presented for SARS-CoV-2 non-immune (NI, *n* = 6), vaccinated HIV-1-uninfected (HU, *n* = 60), vaccinated HIV-1-infected (HIV, *n* = 50), COVID-19 HIV-1-uninfected (COVID HU, *n* = 17), COVID-19 HIV-1-infected (COVID HIV, *n* = 26). The IgG and IgA levels are presented as antibody ratios obtained by dividing the extinction of the sample by that of the assay calibrator. The cut-off ratio for a positive IgG or IgA level at ≥1.1 is indicated by the dashed line. Significance was tested by Wilcoxon signed-rank tests. All vaccinated and COVID-19 convalescent subjects had significantly higher IgG- (*p* ≤ 0.0001) and IgA-levels (*p* ≤ 0.001) compared to the NI group (*p*-value not depicted in the graph). Significance was tested by Mann–Whitney U tests. **** = *p* ≤ 0.0001.

**Figure 3 viruses-14-00651-f003:**
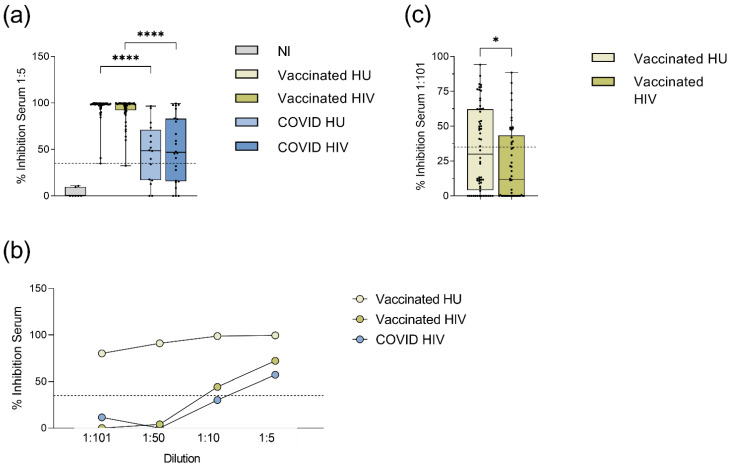
Serum neutralization capacity against SARS-CoV-2 in vaccinated and COVID-19 convalescent donors as measured by a surrogate ELISA neutralization assay. (**a**) Neutralizing activity in serum of the BNT162b2 mRNA vaccinated and the COVID-19 convalescent subjects at a 1:5 serum dilution. All vaccinated and convalescent groups had significantly higher neutralizing activity compared to the NI group (*p* ≤ 0.0006, not depicted in the graph). Medians and IQRs are presented for SARS-CoV-2 non-immune (NI, *n* = 6), vaccinated HIV-1-uninfected (HU, *n* = 57), vaccinated HIV-1-infected (HIV, *n* = 50), COVID-19 HIV-1-uninfected (COVID HU, *n* = 16), COVID-19 HIV-1-infected (COVID HIV, *n* = 24). (**b**) Neutralizing activity of serial dilutions of sera from three subjects with high or low neutralization activity as determined by a 1:5 serum dilution. (**c**) Differences in serum neutralizing capacity between HIV-1-infected (*n* = 50) and -uninfected vaccinated subjects (*n* = 60) at a 1:101 serum dilution. Medians and IQRs are presented. Significance was tested by Mann–Whitney U tests * = *p* ≤ 0.05, **** = *p* ≤ 0.0001. The cut-off for a positive result at ≥35% inhibition is indicated by the dashed lines.

**Figure 4 viruses-14-00651-f004:**
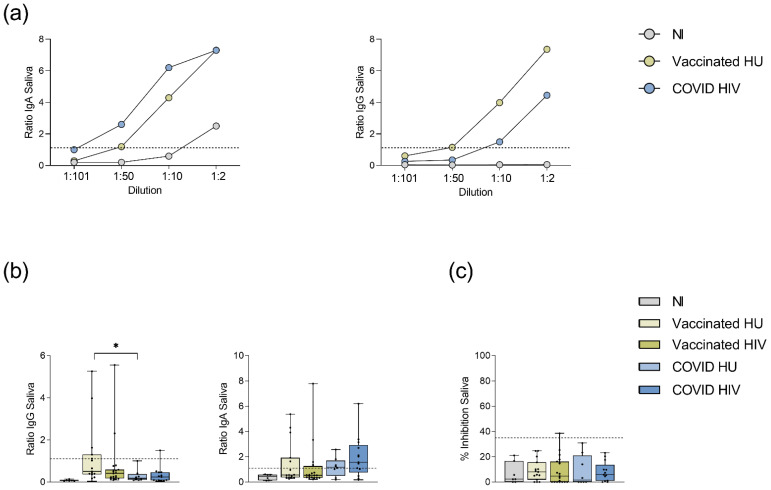
SARS-CoV-2 spike-specific antibodies in saliva. (**a**): Serial titration curves for determination of spike-specific IgA and IgG antibodies in saliva, *n* = 1 each. (**b**): Spike-specific IgG and IgA antibody levels in the saliva of BNT162b2 mRNA vaccinees and COVID-19 convalescent donors given as antibody ratios obtained by dividing the extinction of the sample by that of the calibrator. All vaccinated (*p* ≤ 0.001) and convalescent groups (*p* ≤ 0.030) had significantly higher spike-specific IgG levels in saliva than the NI group. Only convalescent subjects had significantly higher saliva anti-spike IgA levels above the NI group (*p* ≤ 0.029). The cut-off ratio for a positive result at ≥1.1 is indicated by the dashed line. (**c**): Except for one individual, no neutralizing capacity against SARS-CoV-2 could be detected in saliva at a 1:5 dilution. The cut-off for positive neutralization capacity (≥35% inhibition) is indicated by the dashed line. SARS-CoV-2 non-immune (NI, *n* = 6), vaccinated HIV-1-uninfected (HU, *n* = 15), vaccinated HIV-1-infected (HIV, *n* = 19), COVID-19 HIV-1-uninfected (COVID HU, *n* = 8), COVID-19 HIV-1-infected (COVID HIV, *n* = 13). Significance was tested by Mann–Whitney U tests. * = *p* ≤ 0.05. Medians and IQRs are presented.

**Table 1 viruses-14-00651-t001:** Characteristics of the study participants.

Characteristics by SARS-CoV-2 Immunity	HIV-1-Infected (*n* = 82)	HIV-1-Uninfected (*n* = 77)	*p* Value
BNT162b2 mRNA vaccinated (*n* = 110)			
*n*	50	60	
age (years)	55 (46–60)	42 (30–53)	<0.0001
female/male ratio	16/34	28/32	0.171
days post first boost	37 (21–53)	26 (16–68)	0.710
prime-boost interval	42 (23–42)	23 (21–42)	0.0024
CD4 count	634/µL (370–906)	/	/
CD4 count below 300/µL	5/50	/	/
viral load(copies/mL)	<20 (<20−20)	/	/
COVID-19 infected (*n* = 43)			
*n*	26	17	
age (years)	44 (39–57)	48 (35–59)	0.897
female/male ratio	9/17	6/11	>0.999
days post infection	68 (37–140)	72 (22–131)	0.327
CD4 count	551/µL (382–761)	/	/
CD4 count below 300/µL	3/26	/	/
viral load (copies/mL)	<20 (<20–<20)	/	/
SARS-CoV-2 NI (*n* = 6)			
*n*	6	/	/
age (years)	40 (34–56)	/	/
female/male ratio	1/5	/	/
CD4 count	571 (429–921)	/	/
CD4 count below 300/µL	0/6	/	/
viral load (copies/mL)	<20 (<20–30)	/	/

NI = non-immune subjects were SARS-CoV-2 seronegative, SARS-CoV-2 unvaccinated subjects without history of COVID-19. Statistical comparisons between HIV-1-infected and HIV-1-uninfected were performed using Mann–Whitney U-test (age and days post first boost/infection) or Fisher’s exact test (female/male ratio). Indicated numbers for age, CD4 counts, and viral load represent medians with IQR in brackets.

**Table 2 viruses-14-00651-t002:** Multivariate linear regression analysis of the serum spike-specific IgG levels among SARS-CoV-2 vaccinated subjects.

Variable	Estimates (95% CI)	*p* Value
(Intercept)	9.15 (7.41 to 10.88)	<0.0001
Age, per decade	−0.22 (−0.43 to 0.00)	0.048
HIV-1-infected	−0.27 (−0.88 to 0.35)	0.394
Female gender	0.15 (−0.39 to 0.69)	0.581
Time after vaccine, per week	−0.17 (−0.23 to −0.11)	<0.0001
Prime-Boost interval	−0.01 (−0.04 to 0.02)	0.439

**Table 3 viruses-14-00651-t003:** Multivariate linear regression analysis of the serum spike-specific IgG levels of SARS-CoV-2 vaccinated subjects compared to COVID-19 convalescent subjects.

	Vaccinated HU		Vaccinated HIV	
Variable	Estimates (95% CI)	*p* Value	Estimates (95% CI)	*p* Value
(Intercept)	8.29 (7.76 to 8.82)	<0.0001	7.89 (7.35 to 8.42)	<0.0001
Age, per decade	−0.14 (−0.36 to 0.09)	0.242	−0.14 (−0.36 to 0.09)	0.242
COVID HU	−3.44 (−4.41 to −2.48)	<0.0001	−3.04 (−4.03 to −2.05)	<0.0001
COVID HIV	−3.38 (−4.24 to −2.52)	<0.0001	−2.98 (−3.86 to −2.09)	<0.0001
Female gender	0.37 (−0.21 to 0.96)	0.208	0.37 (−0.21 to 0.96)	0.208
Time after vaccine, per week	−0.12 (−0.16 to −0.07)	<0.0001	−0.12 (−0.16 to −0.07)	<0.0001

**Table 4 viruses-14-00651-t004:** Multivariate linear regression analysis of serum spike-specific IgA levels among SARS-CoV-2 vaccinated subjects.

Variable	Estimates (95% CI)	*p* Value
(Intercept)	4.91 (1.63 to 8.19)	0.004
Age, per decade	−0.11 (−0.51 to 0.30)	0.605
HIV-1-infected	−1.21 (−2.37 to −0.05)	0.041
Female gender	−0.47 (−1.49 to 0.54)	0.360
Time after vaccine, per week	−0.30 (−0.41 to −0.19)	<0.0001
Prime-Boost interval	0.04 (−0.02 to 0.09)	0.178

**Table 5 viruses-14-00651-t005:** Multivariate linear regression analysis of the serum spike-specific IgA levels of SARS-CoV-2 vaccinated subjects compared to COVID-19 convalescent subjects.

	Vaccinated HU		Vaccinated HIV	
Variable	Estimates (95% CI)	*p* Value	Estimates (95% CI)	*p* Value
(Intercept)	6.70 (5.90 to 7.50)	<0.0001	7.89 (7.35 to 8.42)	<0.0001
Age, per decade	−0.14 (−0.48 to 0.21)	0.430	−0.14 (−0.48 to 0.21)	0.430
COVID HU	−0.58 (−2.03 to 0.88)	0.432	0.31 (−1.19 to 1.80)	0.686
COVID HIV	0.58 (−1.90 to 0.75)	0.390	0.31 (−1.04 to 1.66)	0.653
Female gender	−0.10 (−0.98 to 0.78)	0.820	−0.10 (−0.98 to 0.78)	0.820
Time after vaccine, per week	−0.22 (−0.29 to −0.15)	<0.0001	7.89 (7.35 to 8.42)	<0.0001

**Table 6 viruses-14-00651-t006:** Multivariate linear regression analysis of serum spike-specific neutralizing antibodies at a 1:101 dilution among the SARS-CoV-2 vaccinated subjects.

Variable	Estimates (95% CI)	*p* Value
(Intercept)	48.11 (17.01 to 79.22)	0.003
Age, per decade	0.24 (−3.61 to 4.10)	0.901
HIV-1-infected	−11.82 (−22.83 to −0.82)	0.036
Female gender	−5.86 (−15.48 to 3.77)	0.230
Time after vaccine, per week	−3.15 (−4.18 to −2.11)	<0.0001
Prime-Boost interval	−0.13 (−0.63 to 0.37)	0.178

**Table 7 viruses-14-00651-t007:** Proportions of donors with measurable anti-spike antibodies in the saliva.

	IgA			IgG		
	Neg.	Bord.	Pos.	Neg.	Bord.	Pos.
SARS-CoV-2 NI (*n* = 6)	6	0	0	6	0	0
Vaccinated HU (*n* = 15)	9	1	5	9	1	5
Vaccinated HIV (*n* = 19)	14	0	5	17	0	2
COVID HU (*n* = 8)	2	1	5	7	1	0
COVID HIV (*n* = 13)	3	1	9	12	0	1
*p* value HU vs. HIV						
Vaccinated		0.707			0.115	
COVID		>0.999			>0.999	
*p* value Vaccinated vs. COVID						
HU		0.183			0.124	
HIV		0.012			>0.999	

Antibody levels for each subject were determined as negative (Neg.), borderline (Bord.) or positive (Pos.). Statistical analysis of antibody prevalence between the HIV-1-uninfected (HU) and the HIV-1-infected (HIV) groups was determined by Fisher´s exact test excluding borderline results. Comparison between SARS-CoV-2 NI and the other groups was significant only for IgA prevalence in the COVID subgroups: COVID HU vs. NI: *p* = 0.021, COVID HIV vs. NI: *p* = 0.009. In convalescent subjects, the spike-specific IgA antibodies were more frequently observed compared to the spike-specific IgG antibodies in both groups (COVID HIV: *p* = 0.001; COVID HU: *p* = 0.0256; Fisher’s exact test).

**Table 8 viruses-14-00651-t008:** Normalized saliva anti-spike IgG and IgA levels.

	*n* IgG^+^	Normalized IgG Level	*n* IgA^+^	Normalized IgA Level
Vaccinated HU (*n* = 15)	5	1.3 (0.7–1.6)	5	1.3 (1.1–2.9)
Vaccinated HIV (*n* = 19)	2	0.7 (0.5–0.9)	5	0.3 (0.2–2.2)
COVID HU (*n* = 8)	0	/	5	1.1 (0.6–1.6)
COVID HIV (*n* = 13)	1	0.2 (0.2–0.2)	9	1.0 (0.7–1.6)

Medians with IQRs in brackets are presented for the spike-specific IgA and IgG levels given as ratios obtained by dividing the extinction of the sample by that of the calibrator and normalizing it to the protein concentration (µg/µL) in saliva. Only values of subjects that had saliva Ig ratios above the cut-off before normalization are shown. HU: HIV-1-uninfected. A ratio of <0.8 was considered as negative, a ratio of 0.8–1.1 as borderline and ≥1.1 as positive. No significant differences as tested by Mann-Whitney U tests.

## Data Availability

The data presented in this study are available on request from the corresponding author.

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
