# Peer review of "Characterization of Serum and Mucosal SARS-CoV-2-Antibodies in HIV-1-Infected Subjects after BNT162b2 mRNA Vaccination or SARS-CoV-2 Infection"

_viruses, 2022, doi:10.3390/v14030651_

Round 1

Reviewer 1 Report

In their study entitled “Characterization of serum and mucosal SARS-CoV-2-antibodies in HIV-1-infected subjects after BNT162b2 mRNA vaccination or SARS-CoV-2 infection”, the authors measured the Spike protein specific antibody responses after BNT162b2 mRNA vaccination or COVID infection in HIV infected and uninfected individuals. Their results showed that the vaccine induced binding and neutralizing antibody responses were lower in HIV-1 infected participants. The observation may provide additional data to characterize the effects of COVID vaccines among people living with HIV, however several concerns need to be carefully addressed.

  1. The dose and schedule of vaccination are missing. And details with regard to participant recruitment procedures, such as the inclusion and exclusion criteria for study participants, should be provided. The status of comorbidity beside HIV-1 infection should be clarified. More importantly, the sample collection time points varied a lot intra- and inter-groups, which might significantly impact the observation of antibody responses.
  2. The sample size of the current study is very small, which makes it hard to do any sub-analysis to clarify the potential influences of age, gender and sampling time on the observed differences between HIV-1 infected and healthy vaccinees.
  3. The immunogenicity and safety of major licensed COVID-19 vaccines (mRNA vaccine, inactivated vaccine and viral vector vaccine) among people living with HIV have been evaluated by previous studies. The scientific merit of this study should be discussed in comparison with the previous studies.
  4. Given that the sampling time varied profoundly and the antibody responses correlated with the sampling time (Supplementary figures), the comparisons between vaccine induced and infection induced antibody responses should be cautiously interpreted.
  5. It will be more reliable if the measurements of IgG/IgA in saliva can be normalized to the total protein concentrations.
  6. One HIV infected individual (ID501) showed extremely low CD4/CD8 cell counts. It may not be appropriate to analyze the data of this participant together with others.

Reviewer 2 Report

In this manuscript, Thomas Harrer and his colleagues investigated the levels of spike-specific IgA and IgG antibodies and neutralizing antibodies against the spike S1/RBD domain in serum and saliva in HIV-1 infected and non-infected people who got BNT162b2 mRNA vaccine or infected with SARS-CoV-2. The results showed that, after two vaccine doses, both HIV-1 positive and negative subjects exhibited anti-spike IgG and IgA antibodies in serum. Furthermore, the levels of these antibodies and neutralizing ability were dramatically higher than SARS-CoV-2 convalescents. However, in comparison to the HIV-1-uninfected group, the HIV-1-infected group displayed less anti-spike IgG and lower neutralizing activity in serum. Anti-spike IgG and IgA antibodies were detectable in both vaccinees and convalescents in saliva but at lower frequencies. The authors also compared their findings and other studies in discussion and explained the differences. In general, the manuscript is well organized, and data are presented clearly and conclusively. I believe the finding that the BNT162b2 mRNA vaccine induces anti-SARS-CoV-2 spike antibodies in HIV-1 patients but lower efficiency than HIV-1-uninfected people will better understand the humoral SARS-CoV-2-specific immune response in HIV-1-infected people.

Author Response

Reviewer 2: In this manuscript, Thomas Harrer and his colleagues investigated the levels of spike-specific IgA and IgG antibodies and neutralizing antibodies against the spike S1/RBD domain in serum and saliva in HIV-1 infected and non-infected people who got BNT162b2 mRNA vaccine or infected with SARS-CoV-2. The results showed that, after two vaccine doses, both HIV-1 positive and negative subjects exhibited anti-spike IgG and IgA antibodies in serum. Furthermore, the levels of these antibodies and neutralizing ability were dramatically higher than SARS-CoV-2 convalescents. However, in comparison to the HIV-1-uninfected group, the HIV-1-infected group displayed less anti-spike IgG and lower neutralizing activity in serum. Anti-spike IgG and IgA antibodies were detectable in both vaccinees and convalescents in saliva but at lower frequencies. The authors also compared their findings and other studies in discussion and explained the differences. In general, the manuscript is well organized, and data are presented clearly and conclusively. I believe the finding that the BNT162b2 mRNA vaccine induces anti-SARS-CoV-2 spike antibodies in HIV-1 patients but lower efficiency than HIV-1-uninfected people will better understand the humoral SARS-CoV-2-specific immune response in HIV-1-infected people.

We thank the reviewer for these positive comments and for the time and efforts to review our manuscript.

Reviewer 3 Report

The manuscript by Schmidt et al. investigates the humoral response in cART-treated HIV-1 infected subjects and HIV-1 uninfected subjects after SARS-CoV2 infection or vaccination with BNT162b2 mRNA. For this study, sera and saliva samples were used to detect IgG and IgA against the SARS-CoV-2 spike protein as well as neutralizing antibodies against the S1/RBD domain of the SARS-CoV-2 spike protein.

After reading the manuscript, I have some suggestions.

Introduction:

The introduction includes the most important things to consider.

Materials and Methods:

-The subjects included in the study are well described but not matched. Differences in the days post first boost/post infection are too evident.

-I recommend to include the IQR of the data.

-Information about how long are HIV infected and how long are they virally suppressed is missing.

Results:

In my opinion, the results part can be improved.

-Line 175: Table 2 can be included in the Supplementary Material because does not provide additional information to Figure 1. The same for Table 3 and Table 5 (Move Table 3 and Table 5 to Supplementary Material).

-Line 138-173: Something striking is that vaccinated subjects have higher IgG titers than infected subjects, even in HIV+. Can be the time point of the sample the reason? Considering the differences observed in the days after vaccination for vaccinated group and after infection for COVID group some differences may be due to this rather than vaccination or infection. The small size of each group does not help to elucidate it.

-Line 291: Due to few positive samples for IgG in the saliva all the statistical significance found by the authors is too weak. The authors should include a comment discussing it. Why they compare in the Figure 4b the different groups with the NI group? Which is the intention of this analysis and why they do it only in this figure? This must be included in the manuscript.

Round 2

Reviewer 1 Report

The authors mentioned in their reply that the sampling time of this study is different from previous observations.  It is known that vaccine elicited antibody responses wane over time. Comparison of antibody responses elicited by an inactivated COVID-19 at ~4 weeks post the 2nd dose showed no significant difference between PLWH and healthy vaccinees (EClinicalMedicine. 2022 Jan;43:101226. doi: 10.1016/j.eclinm.2021.101226.), while comparison at ~40 days post vaccination showed significant lower antibody levels in PLWH (Int Immunopharmacol. 2022 Jan;102:108383. doi: 10.1016/j.intimp.2021.108383.).  I suggest the authors to discuss the potential difference in antibody decline rate between PLWH and healthy individuals and how it may influence their observation, as the time spans after boost are different between HIV infected and healthy people in their study.

Author Response

We thank the reviewer for the time and efforts to improve our manuscript and the very fast review. According to the suggestions of reviewer 1 we adapted the discussion (lines 501 to 509). We discuss the point that a faster decay of antibody responses may account for our observation of lower antibody responses after BNT162b2 mRNA vaccination in HIV-1 infection and that differences in sampling time between groups in our study warrant caution with respect to estimating the precise impact of HIV-1 infection on antibody levels after BNT162b2 mRNA vaccination. In addition, we included the two suggested references.